# Information-Theoretic Guarantees for Recovering Low-Rank Tensors from Symmetric Rank-One Measurements

**Eren C. Kızıldağ**                                                         KIZILDAG@ILLINOIS.EDU

*Department of Statistics, University of Illinois at Urbana-Champaign* [*]

**Editors:** Gautam Kamath and Po-Ling Loh

## Abstract

We investigate the sample complexity of recovering tensors with low symmetric rank from symmetric rank-one measurements, a setting particularly motivated by the study of higher-order interactions in statistics and the analysis of two-layer polynomial neural networks. Using a covering number argument, we analyze the performance of the symmetric rank minimization program and establish near-optimal sample complexity bounds when the underlying distribution is log-concave. Our measurement model involves random symmetric rank-one tensors, leading to involved probability calculations. To address these challenges, we employ the Carbery-Wright inequality, a powerful tool for studying anti-concentration properties of random polynomials, and leverage orthogonal polynomial expansions. Additionally, we provide a sample complexity lower bound via Fano's inequality, and discuss broader implications of our results for two-layer polynomial networks.

**Keywords:** Symmetric tensors, tensor recovery, rank minimization, covering numbers, low-rank, log-concave distributions.

## 1. Extended Abstract

We study the problem of recovering an unknown, order-$\ell$ tensor $\boldsymbol{\mathcal{T}}^* \in \mathbb{R}^{d \times \cdots \times d}$ from random measurements of the form

$$Y_i = \langle \boldsymbol{\mathcal{T}}^*, \boldsymbol{X}_i^{\otimes \ell} \rangle, \quad i = 1, \ldots, N, \tag{1}$$

where $N$ is the sample size and $\boldsymbol{X}_i \in \mathbb{R}^d$ are i.i.d. random vectors with i.i.d. entries sampled from a log-concave distribution on $\mathbb{R}$. We assume $\boldsymbol{\mathcal{T}}^*$ has low symmetric rank, i.e.,

$$\mathrm{rank}_S(\boldsymbol{\mathcal{T}}^*) := \min\{t \geq 1 : \boldsymbol{\mathcal{T}}^* = \textstyle\sum_{i \leq t} \lambda_i \boldsymbol{v}_i^{\otimes \ell}, \lambda_1, \ldots, \lambda_r \in \mathbb{R}, \boldsymbol{v}_1, \ldots, \boldsymbol{v}_r \in \mathbb{R}^d\} \leq r$$

for some $r$. This setting arises naturally in the study of higher-order interactions in statistics (Bien et al., 2013; Basu et al., 2018; Hao et al., 2020), where the unknown tensor often exhibits low-rank structure (Sidiropoulos and Kyrillidis, 2012; Hung et al., 2016; Hao et al., 2020). Moreover, our setting is also closely related to the problem of learning two-layer polynomial neural networks, see, e.g., Soltanolkotabi et al. (2018); Du and Lee (2018); Emschwiller et al. (2020); Sarao Mannelli et al. (2020); Kızıldağ (2022); Martin et al. (2024); Gamarnik et al. (2024).

Our main contributions are summarized as follows:

- *Strong Recovery*: We establish that for $N = \Omega(dr)$, the symmetric rank minimization program, $\min_{\boldsymbol{\mathcal{T}}} \mathrm{rank}_S(\boldsymbol{\mathcal{T}})$ subject to $\langle \boldsymbol{\mathcal{T}}, \boldsymbol{X}_i^{\otimes \ell} \rangle = Y_i, \forall i = 1, \ldots, N$, recovers *all* $\boldsymbol{\mathcal{T}}^*$ with probability one. Our proof leverages multiple techniques, including Carbery-Wright inequality for the anti-concentration of random polynomials (Carbery and Wright, 2001), orthogonal polynomial expansions (Lalley; Szegö, 1939), covering number estimates for low-rank tensors (Zhang and Kileel, 2023), and monotonicity of covering numbers (Vershynin, 2018).

---

[*] Extended abstract. Full version appears as (Kızıldağ, 2025).

- *Sample Complexity Lower Bound*: In a different statistical setting where $\mathcal{T}^*$ is drawn from a discrete space, we establish a sample complexity lower bound: any estimator $\widehat{\mathcal{T}}$ for $\mathcal{T}^*$, whether deterministic or randomized, incurs an estimation error of at least some $\delta > 0$, unless $N = \widetilde{\Omega}(dr^{1-\gamma})$, where $\gamma > 0$ is arbitrary. To prove this result, we establish a packing number bound for symmetric tensors with low symmetric rank, potentially of independent interest, using a variant of the Gilbert-Varshamov lemma from coding theory (Gilbert, 1952; Varshamov, 1957) derived via the probabilistic method (Alon and Spencer, 2016).

- *Implications for Neural Networks*: Consider a two-layer neural network of width $r$, computing $\sum_{1 \le j \le r} a_j^* \sigma(\langle \boldsymbol{W}_j^*, \boldsymbol{X} \rangle)$ on input $\boldsymbol{X} \in \mathbb{R}^d$, where $\boldsymbol{W}_j^* \in \mathbb{R}^d$ and $a_j^* \in \mathbb{R}$ are ground-truth weights, and $\sigma(x) = x^\ell$ is the activation function. Our results provide improved sample complexity bounds in the *underparameterized* regime, $r = O(d^{\ell-1})$; they remain competitive in the *overparameterized* regime, $r = \Omega(d^{\ell-1})$, particularly when $a_i^* = \Theta(1)$ or when the spectral norm of $\boldsymbol{W} \in \mathbb{R}^{r \times d}$ with rows $\boldsymbol{W}_j^*$ grows polynomially with $\max\{r, d\}$.

Our work aligns with a broad literature on low-rank matrix and tensor recovery (Candes and Tao, 2005; Candès et al., 2006; Cai et al., 2010; Eldar et al., 2012; Mu et al., 2014; Rauhut et al., 2017; Cai et al., 2020; Ahmed et al., 2020; Grotheer et al., 2022; Luo and Zhang, 2023). Much of this literature adopts measurement models of form $Y_i = \langle \mathcal{T}^*, \mathcal{X}_i \rangle$ for tensors $\mathcal{X}_i$ consisting of i.i.d. sub-Gaussian entries or satisfying the tensor restricted isometry property, simplifying probability estimates. Our approach extends beyond these settings and captures polynomial networks.

## Acknowledgments

I would like to thank anonymous ALT 2025 referees and the area chair for their valuable feedback, which helped improve the presentation and strengthen the results.

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
