# OpenReview forum: "Information-Theoretic Guarantees for Recovering Low-Rank Tensors from Symmetric Rank-One Measurements"
_algorithmiclearningtheory.org/ALT/2025/Conference — ALT 2025_

### Official Review · Reviewer_KoE3 · 2024-10-30
**Some comments on the generality of the result and the presentation of the paper**

**Rating:** 6
**Confidence:** 3

**Review:**

In this paper the authors study the sample complexity of recovering low rank symmetric tensors from random symmetric rank-one measurements.

**Settings and result:**
Let the rank be $r\,$ the dimension $d\,$ and the tensor order $\ell$.
We are given samples of the form $Y_i=\langle T, X_i^{\otimes \ell}\rangle\,$ where $X_i\sim N(0, I)\.$
The main result of the paper shows that, information theoretically, $O(\ell^2\cdot r\cdot d)$ observations are enough to learn $T\.$
The authors complement this result by showing that $\Omega(d\cdot r^{1-\epsilon}/(\ell\log d))$ samples are necessary.

The problem is closely related to that of learning two-layer polynomial networks.

The result itself is not surprising,  the proof uses a combination of moment analysis for Gaussians (via the corresponding orthonormal base of Hermite polynomials), anti-concentration, and metric entropy bounds.
Given that the techniques used are well-understood, I feel the paper would have been stronger had the Gaussian assumptions been relaxed.
More concretely, the authors do not clarify what are the limits of the technique used and do not try to investigate the distributional properties necessary to obtain a result as Theorem 2.
While there is value in this work,  I am inclined to consider this its main limitation.





**Comments:**
- I don't quite see the point of the section *"Comparison with ERM"*. There the authors show that the sample complexity of the least square estimator is significantly worse. I do not see how this is a meaningful comparison since this estimator is allowed to optimize over the entire $(\mathbb{R}^{d})^{\otimes \ell}$ space rather than in the lower dimensional space given by low rank symmetric tensors. Indeed, by adding such a constraint (via a regularizer or a constraint) the estimator would yield better guarantees (my guess is that these would be more in line with Theorem 2).
A more meaningful comparison could be made against larger classes of efficient estimators.
By employing well-known (conditional) computational lower bounds, the authors could obtain a more insightful version of Theorem 5.
- The paper could be better contextualized. The literature on recovery of low-dimensional structures is vast and I believe a more accurate picture could improve the presentation of the results, as well as provide a starting point to improve Theorem 5.
A comparison with general (and efficient) M-estimators would be useful.
One additional line of work that could be worth looking at is that on oblivious adversarial models, a generalization of matrix\tensor completion.

**Comments on the presentation:**
- The formulation of Theorem $2$ is a bit confusing. $T^*$ is first defined as a specific tensor, then the same symbol is overloaded to represent all tensors satisfying the given linear constraints.
- In *"The Road for Proving Theorem 2"* the authors try to outline the proof structure of their results. I believe the section could be written better. As a concrete example the authors write *"We bound the covering numbers of $\zeta_S(2r)$ using a certain monotonicity property of covering numbers"*.
However, this is quite a natural argument, that is not impossible to convey in a few lines. Hence a more effective strategy would be that of explaining what this property is, why it holds and how it is used here.
- The presentation of the results is also a bit chaotic. For example, I did not find Proposition 3 more insightful than Theorem 2. It appears to me the importance of the proposition lies in the fact that it implies Theorem 2. In this case it might be better to just present Theorem 2 and then in *"The Road for Proving Theorem 2"* argue that the result is obtained via this proposition.

**Questions:**
- What can be said about the quadratic dependency of the sample complexity on $\ell$?

**Paper Award:**

No

---

> ### Author Response · Authors · 2024-11-25
> **Response to Comments**
>
> Thank you very much for your review. Below, we outline the extension beyond Gaussians to log-concave distributions, showing that the sample complexity bound is unchanged. We also address your other comments.
>
> **Gaussianity Assumption**
> We appreciate the suggestion to relax the Gaussian assumption, and agree that doing so strengthens our contributions. As outlined in Remark 8, our results indeed extend to arbitrary log-concave distributions. This is now made explicit through the following argument, which will be incorporated into the final version:
>
> — Carbery-Wright inequality remains valid for log-concave distributions (Theorem 14).
>
> — Gaussianity is used only in Lemma 19, where (34) is calculated via Hermite expansions.
>
> **Generalization to Log-Concave Distributions**
> For an arbitrary log-concave distribution $\mathcal{D}$, the associated family of orthonormal polynomials $(P\_i(x))\_{i\ge 1}$ exist by classical results (Szego [1]). We modify Lemma 19 using the following.
>
> 1.  Let $m\_i:=\mathbb{E}\_{X\sim \mathcal{D}}[X^i]$. Log-concavity ensures all moments $m_i$ are finite [2].
> 2.  Define $D_n$ as the determinant of  $(n+1)\times (n+1)$ Gram matrix with entries $(m\_{i+j})\_{0\le i,j\le n}$. It can be shown that $D\_i>0$ for all $i$ (details will be included in final version).
> 3. Using Proposition 1 in [3], we have
> $$
> \mathbb{E}\_{X\sim\mathcal{D}}[X^n P\_n(X)] = \sqrt{\frac{D\_n}{D\_{n-1}}},\quad \text{for}\quad n\ge 1.
> $$
> Moreover, the proof of Proposition 1 in [3] also implies that $\mathbb{E}[X^k P\_n(X)] = 0$ for $k<n$.
>
> Substituting orthonormal polynomials $\boldsymbol{P}\_{\boldsymbol{\alpha}}(\boldsymbol{X}) = P\_{\alpha_1}(X_1) \cdots P\_{\alpha_d}(X_d)$, Lemma 19 extends to:
> $$
> \mathbb{E}\_{\boldsymbol{X}\sim \mathcal{D}^d}[\boldsymbol{X}\_{\mathcal{I}}\boldsymbol{P}\_{\boldsymbol{\alpha}}(\boldsymbol{X})]= \mathbf{1}_\{\boldsymbol{\beta}\_{\mathcal{I}} = \boldsymbol{\alpha}\}\cdot \prod\_{1\le j\le d} \sqrt{\frac{D\_{\alpha\_j}}{D\_{\alpha\_{j}-1}}}.
> $$
> (here, $D\_{\alpha_j}/D\_{\alpha_j-1}:=1$ if $\alpha\_j=0$).
>
> This immediately generalizes Proposition 15 (by replacing $\widehat{\boldsymbol{H}}\_{\boldsymbol{\alpha}}$ with $\boldsymbol{P}\_{\boldsymbol{\alpha}}$ and slightly modifying (37)-(39)):
> $$
> \mathbb{E}\_{\boldsymbol{X}\sim \mathcal{D}^d}[\langle \boldsymbol{\mathcal{T}} , \boldsymbol{X}^{\otimes \ell}\rangle] \ge \Xi \cdot ||\boldsymbol{\mathcal{T}}|| \_F^2,\quad\text{where}\quad \Xi:= \mathcal{C}(\mathcal{D},\ell)^d \quad\text{and}\quad \mathcal{C}(\mathcal{D},\ell) = \frac{\min\_{0\le i\le \ell} D\_i}{\max\_{0\le i\le \ell}D\_i}.
> $$
> Here, $\mathcal{C}(\mathcal{D},\ell)$ is positive, finite, and it depends only on the moments $m_i$.
>
> With this, the proof of Proposition 3 when $\boldsymbol{X}\sim \mathcal{D}^d$ is identical, except that $\ell!$ in (16) and (24) is replaced with $\Xi$.  Importantly, since $\Xi$ is independent of $\epsilon$, the sample bound $O(\ell^2 \cdot d \cdot r)$ remains unchanged.
>
> By broadening our results to log-concave distributions—a rich class that includes uniform, exponential, Laplace, and Gamma distributions among many more [12]—we significantly enhance the generality and impact of our theoretical contribution. We believe this extension is essentially the best possible within our framework, as orthogonal expansions for non-product measures and Carbery-Wright type inequalities beyond log-concave distributions are generally less well-understood [13].
>
>
> **Comparison with ERM and Contextualization**
>
> We completely agree with the reviewer regarding the ERM and further contextualization. The current ERM analysis was included to highlight a sharp contrast between structured and unstructured recovery, showcasing how ignoring structure dramatically increases the sample complexity. We will refine this section in the final version by discussing regularized variants of ERM, as well as adding a comparison with M-estimators, and expanding the discussion on low-rank recovery.
>
> As for oblivious adversarial models, we thank the reviewer for bringing this interesting line of work into our attention. We will include a brief discussion on these models, along with references to relevant literature (e.g., [4]-[9]) in the final version.

---

> > ### Author Response · Authors · 2024-11-25
> > **Response to Comments (Continued)**
> >
> > **Presentation**
> > We acknowledge the reviewer’s helpful feedback on presentation and contextualization. In the revisions, we will fix all notational issues, improving the clarity of section “The Road for Proving Theorem 2”. Moreover, we will move Proposition 3 into section “The Road for Proving Theorem 2” and improve the overall flow.
> >
> > **Question on Sample Complexity**
> > Thank you for raising this insightful question. When proving Proposition 3, two terms play a crucial role:  the covering upper bound (15) and the probability estimate (16). For our argument to work, the term $\frac{N}{\ell}\log\frac{1}{\epsilon}$, obtained from (16), must dominate $2r\ell d\log \frac1\epsilon$ from (15). This leads to a quadratic dependence in $\ell$. The $1/\ell$ scaling in $\frac{N}{\ell}\log\frac{1}{\epsilon}$ originates from (16) and appears intrinsic to the Carbery-Wright bound (e.g., $\mathbb{P}[|P(Z)| \le \epsilon]$ indeed scales like $O\_\epsilon(\epsilon^{1/\ell})$ for $P(x) = x^\ell$ and $Z\sim \mathcal{N}(0,1)$). For the covering bound, we are unsure if the dependence of (15) and Lemma 13 on $\ell$ is optimal, a point we will remark in the revisions.
> >
> > We hope our clarifications have addressed your concerns, particularly regarding the extension beyond Gaussians to arbitrary log-concave distributions.  This rich class is fundamental in statistics and machine learning [10,11]; it contains distributions such as uniform, normal, exponential, Laplace, and Gamma among others [12]. This extension not only preserves the sample complexity bound but also broadens the scope of our result, strengthening our theoretical impact.
> >
> > We would deeply appreciate your consideration of these contributions in your evaluation. We thank you for your time.
> >
> > Best wishes,
> >
> > **References:**
> >
> > [1] Gabor Szego, “Orthogonal Polynomials”
> >
> > [2] Sergey G. Bobkov, Gennadiy P. Chistyakov. "On Concentration Functions of Random Variables"
> >
> > [3] Steven Lalley. “Orthogonal Polynomials” Accessed at: https://galton.uchicago.edu/~lalley/Courses/386/OrthogonalPolynomials.pdf
> >
> > [4] Emmanuel J. Candes, Xiaodong Li, Yi Ma, John Wright "Robust Principal Component Analysis?"
> >
> > [5] Kush Bhatia, Prateek Jain, Parameswaran Kamalaruban, Purushottam Kar. "Consistent Robust Regression"
> >
> > [6] Arun Sai Suggala, Kush Bhatia, Pradeep Ravikumar, Prateek Jain. "Adaptive Hard Thresholding for Near-optimal Consistent Robust Regression"
> >
> > [7] Tommaso d'Orsi, Gleb Novikov, David Steurer. "Consistent regression when oblivious outliers overwhelm"
> >
> > [8] Tommaso d'Orsi, Chih-Hung Liu, Rajai Nasser, Gleb Novikov, David Steurer, Stefan Tiegel. "Consistent Estimation for PCA and Sparse Regression with Oblivious Outliers"
> >
> > [9] Ankit Pensia, Varun Jog, Po-Ling Loh. "Robust regression with covariate filtering: Heavy tails and adversarial contamination"
> >
> > [10] Richard J. Samworth. "Recent Progress in Log-Concave Density Estimation"
> >
> > [11] Guenther Walther. "Inference and Modeling with Log-concave Distributions"
> >
> > [12] Mark Bagnoli, Ted Bergstrom. "Log-concave probability and its applications"
> >
> > [13] Itay Glazer, Dan Mikulincer. “Anti-concentration of polynomials: dimension-free covariance bounds and decay of Fourier coefficients”

---

### Official Review · Reviewer_QKj5 · 2024-11-05
**Information-theoretical results on low-rank tensor recovery**

**Rating:** 7
**Confidence:** 4

**Review:**

**Summary**:
This paper studied low-rank symmetric tensor recovery from rank-1 Gaussian measurements. This problem is a tensor generalization of the low-rank matrix recovery problem from rank-1 measurements. This problem also connects to learning a two-layer network with polynomial activation functions, where a teacher network can be written as a low-rank symmetric tensor.

The author considered an NP-hard problem, i.e., rank minimization, and showed that when the number of measurements is $\Omega(rd)$, the rank-minimization program can recover the ground truth with probability 1. This is near-optimal since the number of parameters in a rank-$r$ low-rank tensor is of order $rd$.

The authors also show that empirical risk minimization can recover the ground truth with $\binom{d+\ell-1}{\ell}$ many samples and below $\binom{d+\ell-1}{\ell}$ many samples there are solutions of the ERM that generalize poorly. This is not very surprising since this is the interpolation threshold.

A sample complexity lower bound is also proved for a different statistical model where the ground truth tensor is generated from a discrete space. It is shown that any estimator can not recover the ground truth uniquely with $O(dr^{1-\gamma})$ many samples with $\gamma>0$.

**Pros**:
The paper is well written and the proof combines tools from high dimensional probability, orthogonal polynomials, and information theory. This topic could be interesting in both tensor methods and the neural network community. As discussed in Section 2.4, the sample complexity result has no norm dependence on the weight matrices and it captures the better sample complexity when the width of the network is relatively small (under-parameterized).

The proof combines recent advances in the covering number estimates of low CP-rank tensor space, Carbery-Wright inequality, and Hermite polynomials. This could be an interesting technique for other problems.

**Cons**:
My major concern is that the main result is only a statistical complexity since rank minimization is not computationally feasible. Even the convex relaxation of rank-minimization (nuclear norm minimization) for tensor is NP-hard. So I would interpret this work as Information-theoretical results on low-rank tensor recovery.  It would be more interesting to obtain a sample complexity bound for a computationally efficient algorithm.

**Minor comments**:

1. Page 3 below equation (7): the discussion that "nuclear-norm minimization" is computationally efficient is not correct. It is shown in (Hillar and Lim 13) that nuclear norm minimization is NP-hard as it is the dual norm of the spectral norm.

2. Page 4, Theorem 5. I wonder if there is an error in the threshold case $N=N^*(d,\ell)$ as both case (a) and (b) cover the interpolation threshold and there is an obvious contradiction.

**Paper Award:**

No

---

> ### Author Response · Authors · 2024-11-25
> **Response to Comments**
>
> Thank you very much for your review. Below we address your comment on the information-theoretical nature of our results, as well as your minor comments.
>
> **NP-Hardness & Information-Theoretical Results:**
> We completely agree that deriving sample complexity bounds for computationally efficient methods is an exciting and important future direction. To better position our work and reflect our focus, we are happy to update the title to “Information-Theoretical Guarantees for Low-Rank Tensor Recovery from Symmetric Rank-One Measurements” in the final version.
>
> **Literature Information-Theoretical Results**
> Information-theoretical results serve as a foundational step towards developing efficient methods. We draw a parallel with Wainwright’s work on sparse recovery. He gives a sample complexity bound for LASSO in [1]. In [2], he studies the information-theoretical thresholds of the same model, where he explicitly mentions “This analysis of fundamental limits complements our previous work on sharp thresholds for support set recovery over the same set of random measurement ensembles using the polynomial-time Lasso method”.  In fact, many influential papers in related statistical settings focus exclusively on such guarantees (e.g., [3]-[8]).
>
> Our work builds on this tradition, aiming to provide theoretical insights and benchmarks for a challenging recovery problem. This alignment with foundational contributions was a key motivation for submitting to ALT, where we believe our work resonates with the venue’s theoretical scope.
>
> As you also kindly noted, establishing tight information-theoretical results in our setting is technically challenging, requiring the Carbery-Wright inequality, orthogonal polynomials, and recent advances in the covering number estimates of low CP-rank tensor space. This contrasts with related papers on low-rank tensor recovery from purely i.i.d. measurements, where such results are significantly simpler to obtain (e.g., Eldar et al. (2012), Mu et al. (2014)). Our sharp guarantees provide benchmarks that can guide and inform the development of practical algorithms in future work. We thus lay the foundations for developing efficient algorithms using optimal statistical resources. For instance, our results identify that an algorithm requiring $\Theta(dr)$ samples is sample-optimal, whereas one needing significantly more than $dr$ samples may not be sample-optimal and could potentially be improved.
>
> In the final version, we will include a detailed discussion on convex relaxations and computationally efficient methods, highlighting them as exciting avenues for future work. We hope that this discussion, coupled with our tight theoretical guarantees, will spur further interest in this problem within the ALT community.
>
>
>
> **Neural Networks**
> Thank you very much for your detailed and encouraging remarks regarding the implications of our results for neural networks. As you kindly highlighted, our bounds are relevant not only to the tensor recovery but also to the neural network community, offering sharper sample complexity guarantees without dependence on weight norms.
>
> We would like to further draw your attention to a recent ALT 2024 paper by Daniely and Granot [9], which establishes sample complexity bounds for certain two-layer neural networks (that depend on the norms of weights). This paper reinforces your point, highlighting the interest in sample complexity results for neural networks within ALT’s scope.
>
> **Minor Comments**
> Thank you for pointing these out—we greatly appreciate your careful review. In the final version, we will: (1) explicitly state that nuclear-norm minimization for tensors is NP-hard, and (2) correct Part (b) of Theorem 5 to $N<N^*(d,\ell)$.
>
> We hope our responses and planned revisions address your concerns and further clarify the relevance of our results. We are grateful for your thoughtful feedback and hope that you might consider revising your score in light of our responses. Thank you again for your time.
>
> Best wishes,

---

> > ### Author Response · Authors · 2024-11-25
> > **Reference List**
> >
> > **References:**
> >
> > [1] Martin J. Wainwright. “Sharp Thresholds for High-Dimensional and Noisy Sparsity Recovery Using $\ell\_1$-Constrained Quadratic Programming (Lasso)”
> >
> > [2] Martin J. Wainwright. “Information-theoretic limits on sparsity recovery in the high-dimensional and noisy setting”
> >
> > [3] Vincent Y. F. Tan, Laura Balzano, Stark C. Draper. “Rank Minimization Over Finite Fields: Fundamental Limits and Coding-Theoretic Interpretations”
> >
> > [4] Yonina Eldar, Deanna Needell, Yaniv Plan.  “Uniqueness conditions for low-rank matrix recovery“
> >
> > [5] Zhiqiang Xu. The minimal measurement number for low-rank matrix recovery
> >
> > [6] Wei Wang, Martin J. Wainwright, Kannan Ramchandran. “Information-Theoretic Limits on Sparse Signal Recovery: Dense versus Sparse Measurement Matrices”
> >
> > [7] Jess Banks, Cristopher Moore, Roman Vershynin, Nicolas Verzelen, Jiaming Xu. “Information-Theoretic Bounds and Phase Transitions in Clustering, Sparse PCA, and Submatrix Localization”
> >
> > [8] Yuchen Zhang, John Duchi, Michael I. Jordan, Martin J. Wainwright. “Information-theoretic lower bounds for distributed statistical estimation with communication constraints”
> >
> > [9] Amit Daniely, Elad Granot. "On the Sample Complexity of Two-Layer Networks: Lipschitz Vs. Element-Wise Lipschitz Activation"

---

### Official Review · Reviewer_Hw9h · 2024-11-08

**Rating:** 5
**Confidence:** 3

**Review:**

This paper examines the sample complexity required for recovering low-rank symmetric tensors from symmetric rank-one measurements. The authors adopt a random setting where the rank-one measurements are inner products with rank-one symmetric Gaussian random tensors. By leveraging tools such as covering number arguments and the Carbery-Wright inequality, the paper establishes both upper and lower bounds on the sample complexity.

The nice thing about this paper is that the obtained upper and lower bounds only differs by a polynomial factor with arbitrarily small degree. Additionally, the authors also discuss the applications of their results to two-layer polynomial neural networks. However, the proposed natural rank minimization problem is NP-hard (as noted by the authors), and hence not applicable in real practice. I would be more interested to see a more in-depth analysis of sample complexity of empirical risk minimization, or some convex relaxations of the original optimization problem. Further, this paper only considers the setting where the linear measurements are exactly known. It would be better to extend the analysis to settings where measurement errors are allowed, and explore the dependence of sample complexity on the errors.

**Paper Award:**

No

---

> ### Author Response · Authors · 2024-11-25
> **Response to Comments**
>
> Thank you very much for your review. Below, we address your comments.
>
> **NP-Hardness**
> We fully agree that exploring practical methods, such as convex relaxations of rank minimization, is an exciting and important future direction. In the final version, we will explicitly mention this, as we believe it is a topic of strong interest to the ALT community.
>
> > I would be more interested to see a more in-depth analysis of sample complexity of empirical risk minimization
>
> Regarding ERM, Theorem 5 actually identifies the necessary and sufficient number of samples (interpolation threshold), as well as the generalization performance of solutions to ERM. In the final version, we will expand the discussion surrounding Theorem 5.
>
> Our work provides tight theoretical guarantees that serve as foundational benchmarks for evaluating and guiding future practical methods. For instance, an algorithm requiring $\Theta(dr)$ samples is sample-optimal, whereas one needing significantly more than $dr$ samples may not be sample-optimal and could potentially be improved. This can directly inform algorithm designers, guiding them to the most optimal use of statistical resources.
>
> A parallel can be drawn to Wainwright’s work on sparse recovery. In [1], he gives a sample complexity bound for the LASSO. In [2], he studies the information-theoretical thresholds of the same problem, where he explicitly mentions that “This analysis of fundamental limits complements our previous work on sharp thresholds for support set recovery over the same set of random measurement ensembles using the polynomial-time Lasso method”. Similarly, many important papers in the field focus exclusively on information-theoretic guarantees in statistical settings (e.g., [3]-[8]).
>
> In our case, even deriving tight bounds for rank minimization in tensor recovery involves significant technical work, such as leveraging the Carbery-Wright inequality, orthogonal polynomials, and recent results on certain covering number estimates. This contrasts with related settings (e.g., Eldar et al. (2012), Mu et al. (2014)), where deriving guarantees for rank minimization is considerably simpler. In light of this, our results lay groundwork for future work on convex relaxations, which we will explicitly emphasize in the final version.
>
> **Updating Title** To better position our work, we will update the title to “Information-Theoretical Guarantees for Low-Rank Tensor Recovery from Symmetric Rank-One Measurements” in the final version.
>
>
> **Measurement Errors**
>
> > Further, this paper only considers the setting where the linear measurements are exactly known. It would be better to extend the analysis to settings where measurement errors are allowed, and explore the dependence of sample complexity on the errors.
>
> We echo the reviewer’s suggestion that allowing measurement errors in our model is an important future direction. That said, noiseless models, where measurements are exactly known, are foundational in the literature on recovering low-rank structures. These models provide clean theoretical benchmarks that help identify fundamental limits and guide the development of practical algorithms. In fact, many important papers in the area have focused exclusively on noiseless models (e.g., [9]-[14]). By following this tradition, our work contributes towards a rigorous theoretical foundation for recovering tensors with low symmetric rank. We hope these results will serve as a basis for future extensions to noisy models. We will explicitly highlight this rationale in the final version.
>
>
> **Neural Networks** As you kindly noted, our results have implications for certain neural networks, yielding better sample bounds with no assumptions on the norms of weights. We would like to emphasize a recent ALT 2024 paper by Daniely and Granot [15], which establishes sample complexity bounds for two-layer neural networks. This paper underscores the importance of theoretical guarantees in learning problems, further demonstrating the relevance of our results to both tensor recovery and the neural network community.
>
>
> We hope our explanations address your concerns and clarify how our results contribute to the foundational understanding of low-rank tensor recovery, establish benchmarks for practical algorithms, and connect to fields such as neural networks. These contributions align closely with ALT’s theoretical focus, and we are excited about their potential to inspire future research.
>
> If our responses have clarified your concerns, we would kindly ask you to consider revising your score. Thank you again for your time.
>
> Best regards,

---

> > ### Author Response · Authors · 2024-11-25
> > **Reference List**
> >
> > **References:**
> >
> > [1] Martin J. Wainwright. “Sharp Thresholds for High-Dimensional and Noisy Sparsity Recovery Using $\ell\_1$-Constrained Quadratic Programming (Lasso)”
> >
> > [2] Martin J. Wainwright. “Information-theoretic limits on sparsity recovery in the high-dimensional and noisy setting”
> >
> > [3] Vincent Y. F. Tan, Laura Balzano, Stark C. Draper. “Rank Minimization Over Finite Fields: Fundamental Limits and Coding-Theoretic Interpretations”
> >
> > [4] Yonina Eldar, Deanna Needell, Yaniv Plan.  “Uniqueness conditions for low-rank matrix recovery“
> >
> > [5] Zhiqiang Xu. “The minimal measurement number for low-rank matrix recovery”
> >
> > [6] Wei Wang, Martin J. Wainwright, Kannan Ramchandran. “Information-Theoretic Limits on Sparse Signal Recovery: Dense versus Sparse Measurement Matrices”
> >
> > [7] Jess Banks, Cristopher Moore, Roman Vershynin, Nicolas Verzelen, Jiaming Xu. “Information-Theoretic Bounds and Phase Transitions in Clustering, Sparse PCA, and Submatrix Localization”
> >
> > [8] Yuchen Zhang, John Duchi, Michael I. Jordan, Martin J. Wainwright. “Information-theoretic lower bounds for distributed statistical estimation with communication constraints”
> >
> > [9] Shirin Jalali. “Toward Theoretically Founded Learning-Based Compressed Sensing”
> >
> > [10] Shirin Jalali, H.Vincent Poor. "Universal Compressed Sensing for Almost Lossless Recovery"
> >
> > [11] Alia Abbara, Antoine Baker, Florent Krzakala, Lenka Zdeborová. "On the universality of noiseless linear estimation with respect to the measurement matrix"
> >
> > [12] Erwin Riegler, David Stotz,  Helmut Bolcskei. "Information-Theoretic Limits of Matrix Completion"
> >
> > [13] Raghunandan H. Keshavan , Andrea Montanari, Sewoong Oh."Matrix Completion from a Few Entries"
> >
> > [14] Yihong Wu, Sergio Verdú. "Rényi Information Dimension: Fundamental Limits of Almost Lossless Analog Compression"
> >
> > [15] Amit Daniely, Elad Granot. "On the Sample Complexity of Two-Layer Networks: Lipschitz Vs. Element-Wise Lipschitz Activation"

---

### Author Rebuttal · Authors · 2024-11-25

We thank the reviewers for their time and detailed feedback. In the final version, we will address the reviewers’ suggestions and incorporate edits we commit to below.

We now provide our individual responses.

---

### Meta-Review · Area_Chair_oKcm · 2024-12-09

**Recommendation:** Accept
**Confidence:** 4

**Metareview:**

The paper studies sample complexity of recovering a low rank symmetric tensor from rank-one linear measurements in a random-design setting, giving both upper bounds and lower bounds, which disagree by an arbitrarily small polynomial factor O(dr) vs O(dr^{1-\gamma}).

On the positive side, the results are interesting and the connection to learning two-layer neural networks broadens the audience for the paper. The proof technique is also interesting and perhaps relevant in other settings.

The main shortcoming of the paper is that the estimator studied is not computationally tractable, and hence the results should be interpreted as information-theoretic only. However, we view this as interesting progress and thus recommend acceptance.

**Paper Award:**

No